# Strain Rate Dependent Behavior of Vinyl Nitrile Helmet Foam in Compression and Combined Compression and Shear

**Nicolas Bailly [1,2,\*], Yvan Petit [1,2], Jean-Michel Desrosier [1,2], Olivier Laperriere [1,2], Simon Langlois [3] and Eric Wagnac [1,2]**

[1] Department of Mechanical Engineering, École de Technologie Supérieure, 1100 Notre-Dame Street West, Montréal, QC H3C 1K3, Canada; Yvan.Petit@etsmtl.ca (Y.P.); jean-michel.desrosiers.1@ens.etsmtl.ca (J.-M.D.); olivier.laperriere.1@ens.etsmtl.ca (O.L.); eric.wagnac@etsmtl.ca (E.W.)

[2] Research Center, Department of Traumatology and Acute Care, Hôpital du Sacré-Cœur de Montréal, 5400 Gouin Blvd, Montréal, QC H4J 1C5, Canada

[3] CCM, 3400 Raymond-Lasnier St., Montreal, QC H4R 3L3, Canada; simlanglois8@gmail.com

\* Correspondence: nicolas1.bailly@gmail.com; Tel.: +1-(514)-567-8296



**Featured Application: Results of this work can be used to compare VN foams to other helmet liner materials and could help in the design of new protective devices.**

**Abstract:** Vinyl nitrile foams are polymeric closed-cell foam commonly used for energy absorption in helmets. However, their impact behavior has never been described in isolation. This study aims to characterize the strain rate dependent behavior of three VN foams in compression and combined compression and shear. Vinyl nitrile samples of density 97.5, 125, and 183 kg/m$^3$ were submitted to quasi-static compression (0.01 s$^{-1}$) and impacts in compression and combined compression and shear (loading direction of 45°). For impacts, a drop test rig was used, and a method was developed to account for strain rate variation during impactor deceleration. Young's modulus and stress at plateau were correlated with foam density in both compression and combined loading. Vinyl nitrile foams were strain rate dependent: The absorbed energy at the onset of densification was two to four times higher at 100 s$^{-1}$ than at 0.01 s$^{-1}$. In combined loading, the compressive stress at yield was reduced by 43% at a high strain rate. Compared to expanded polypropylene, vinyl nitrile foams transmitted less stress at the onset of densification for equivalent absorbed energy and presented a larger ratio between the compression and shear stresses in combined loading (0.37 at yield). This larger ratio between the compression and shear stresses might explain why vinyl nitrile helmet liners are thought to be better at reducing head rotational acceleration than expanded polypropylene helmet liners.

**Keywords:** polymeric foam; impact; compression and shear; helmet

## 1. Introduction

Vinyl nitrile (VN) foams are polymeric closed-cell foams, commonly used for energy absorption in helmets [1–4]. These foams are often chosen for their capacity to sustain several impacts without compromising their energy absorption capability. Additionally, it was shown that a helmet with a VN foam liner was better at reducing rotational acceleration than a helmet with an expanded polypropylene (EPP) foam, the most commonly used foam [3]. However, the stress vs. strain behavior of VN foam during impact has never been described.

The mechanical characterization of foam for impact protection is usually performed in pure compression [5–8]. However, typical impacts sustained by helmets are oblique [9–12], loading the

liner material in combined compression and shear. Methods have been developed to test material for energy absorption in combined compression-shear loading [13–18]. For instance, Mosley et al. [18] tested the effect of the anisotropy of polymer foam on the energy absorption in quasi-static loading with two independent displacement actuators, applying, respectively, shear and compression to the sample. However, few studies have characterized impact protective material during high-speed combined compression-shear loading. Among these studies, two methods were used. Hou et al. [19] modified the Split Hopkinson pressure bar to apply high-speed compression and shear loading to honeycombs. This method provided the overall behavior of a structure at a high constant strain rate under combined shear-compression, but the normal and shear behaviors of the structure could not be isolated [19]. To get the normal and shear behaviors separately, Mills and Gilchrist [20], Hong et al. [21], and later Ling et al. [22] used a multiaxial load cell. They developed drop test rigs [20,22] and a linear impactor [21] for impacting foam material in both compression and shear to study polypropylene, expanded polystyrene foam, and aluminum honeycomb.

The mechanical behavior of polymeric foams typically varies with strain rate [23,24]. In a helmet, the foam must deform to absorb impacts of various speeds and energies [23]. Hence, the strain rate dependency of protective foams must be characterized. This is often performed with the Split Hopkinson pressure bar, enabling materials testing at a high constant strain rate [24]. When the Split Hopkinson pressure bar is not applicable (i.e., large samples, or study of compression and shear behaviors in combined loading), researchers have used drop test rigs, where the deformation of the sample is induced by the impact of a falling mass [5,20,22]. During the impact, the impactor decelerates, and the strain rate varies. Impactor deceleration depends on its mass and initial speed, as well as on foam geometrical and mechanical properties [5]. The varying strain rate during the impact not only makes foam behavior difficult to compare, but also to model, as most finite element analysis solvers use material laws with constant strain rates. To alleviate such limitations, Krundaeva et al. [8] combined multiple impact test results to define stress vs. strain curves at constant strain rates. However, this method has never been applied to foam characterization in combined compression and shear.

The objective of this study was to characterize the strain rate dependent behavior of three VN foams in both compression and combined compression and shear. This characterization was performed under quasi-static (compression-only) and impact loading conditions (compression and combined compression and shear). For impacts, a drop test rig was used to study the normal and shear behaviors of the foam separately, and a method was developed based on Krundaeva et al. [8] to account for strain rate variation during the impactor deceleration.

## 2. Materials and Methods

### 2.1. Foam Samples

Three VN closed-cell foams of density ($\rho^*$) 97.5, 125, and 183 kg/m$^3$ were tested. These foams were sourced from Der-Tex (Saco, ME, USA), under the trade name IMPAX VN600, IMPAX VN740, and IMPAX VN1000, respectively. In the manuscript, the three foams are referred to as VN-A, VN-B, and VN-C, respectively. According to the manufacturer, the VN foams were made of nitrile butyl rubber and poly vinyl chloride (NBR-PVC) and were produced using a hot press foaming process with a relative density (R) ranging between 0.09 and 0.11 for VN-A and VN-B and between 0.17 and 0.20 for VN-C. Foams were received as sheets ($500 \times 500 \times 20$ mm$^3$) and then cut into $60 \times 60 \times 20$ mm$^3$ rectangular samples with a bandsaw. The samples were 20 mm thick to correspond to the typical foam thickness in a VN hockey helmet. The samples were weighed, and the density of the foam was verified. Speckle patterns (Halfords Matt Back or White Primer) were applied to one face of the samples for digital image correlation (DIC). A new sample was used for every impact. All tests were performed at room temperature (20 °C, relative humidity between 10 and 30%).

### 2.2. Quasi-Static Compression

Quasi-static compression tests were performed with an electromechanical universal testing machine, Instron 3300 (Instron, Norwood, ME, USA) equipped with the static load cell (2530-10KN, Instron, Norwood, ME, USA) (capacity: 10 kN, accuracy: 0.6%). Samples were compressed between two steel plates at a constant velocity of 0.2 mm/s (strain rate of 0.01 s$^{-1}$) up to 80% compression. Force, time and displacement were recorded at a sampling rate of 1 Hz. The face of each sample with the speckle pattern was filmed using a MotionBLITZ EoSens® mini camera (MIKROTRON, Unterschleissheim, Germany) (4 Hz) to allow for a full-field strain measurement using digital image correlation (DIC), as per previous work [25–27]. Video footage was analyzed with the GOM Correlate (Gom, Braunschweig, Germany) DIC software to obtain lateral and axial local strains (supplementary material Figure S1 and Video 1). Poisson's ratio was obtained by multiplying −1 by the slope of the linear least square fitting of the curve lateral strain vs. axial strain between 0 and 2% compression.

### 2.3. Impact Tests in Compression and Combined Compression and Shear

Impact tests were performed on a free fall guided impact machine (1000_00_MIMAT, Cadex, St-Jean-sur-Richelieu, QC, Canada) equipped with a 6-axis load cell (MC6-2000, AMTI, Watertown, MA, USA) (capacity z-axis: 9 kN, sampling rate 100 kHz). The compression setup consisted of a 5.7 kg free-falling flat mass dropped onto the foam sample positioned horizontally on the load cell (Figure 1a). The combined compression and shear setup was similar to the one described by Mills and Gilchrist [20]. Two identical foam samples were glued with adhesive transfer tape on two surfaces facing each other and having a 45° angle with respect to the horizontal plane. Both foams were simultaneously impacted by a V-shape impactor of 6.3 kg. The surface of the impactor was covered with sandpaper (grit 80) to limit relative motion at the foam-impactor interface. The load cell was positioned under one of the two impacted samples (Figure 1b).

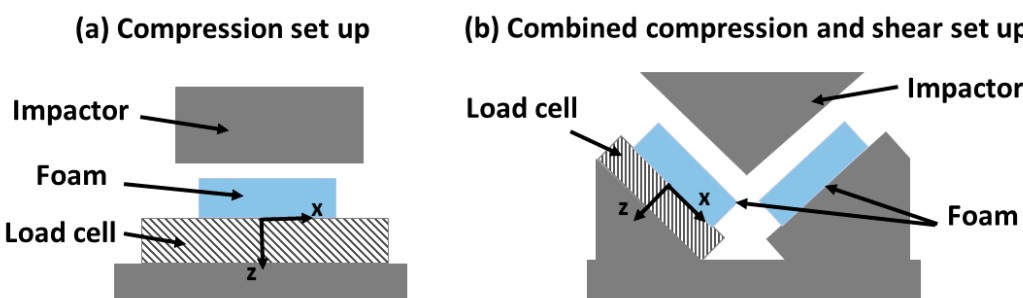

**Figure 1.** Experimental setup of the impact tests in (**a**) compression, (**b**) combined compression and shear.

In both setups, the impactors were dropped from multiple heights starting at 200 mm, up to the maximal height by increments of 200 mm. The maximal height corresponded either to 2.6 m or to the height at which the load reached the maximal load cell capacity (9 kN). The resulting impact speed ranged from 2–7 m.s$^{-1}$, corresponding to typical impact speeds applied to a helmet during a football game, a hockey game or a ski accident [9,12,28]. All impacts were filmed with a high-speed camera MotionBLITZ EoSens® mini (MIKROTRON, Unterschleissheim, Germany) at a sampling rate of 10 kHz. The displacement of the impactor was measured in the footage by automatically tracking markers placed on the impactor with digital image correlation using GOM Correlate. The synchronization of the load cell measurement and the video footage was done by matching the time of the first load measurement above 50 N with the time of the first video footage showing a contact between the impactor and the foam. High-speed video frame was also used to ensure that there was no slippage between the foam and both the laying surface and the impacting surface.

### 2.4. Data Analysis

For quasi-static and impact compressions, force (F), impactor displacement (disp.) and time (t) were converted into engineering strain ($\varepsilon$) (ratio of displacement to initial sample height), strain rate ($\tau$) and engineering stress ($\sigma$) (ratio of force to initial sample area).

Two stress vs. strain curves were obtained under combined compression and shear: A compressive stress vs. strain curve and a shear stress vs. strain curve. The compressive strain and shear strain were calculated using Equation (1), and they were equal because the loading angle was 45°. The compressive stress is the force obtained from the *z*-axis of the load cell (perpendicular to the impacted surface of the foam) divided by the sample area (Figure 1). The shear stress is the force obtained from the *x*-axis of the load cell (along the impacted surface of the foam) divided by the sample area.

$$\varepsilon = \frac{\text{Disp.} \times \frac{\sqrt{2}}{2}}{Sample\ height} \tag{1}$$

During the drop test, the impacting mass decelerated as it compressed the foam, so the strain rate was not constant. The objective of the following analysis, adapted from Kundaeva et al. [8], was to create engineering stress vs. strain curves at a constant strain rate. Those curves at a constant strain rate simplify the comparison between foams. For each foam and testing condition, $\varepsilon$, $\tau$, and $\sigma$ were plotted in a 3D space (Figure 2a). A polynomial surface was fitted with the least absolute residual method using the *fit* function in Matlab R2018b (MatWorks, Natick, MA, USA) [29] (Figure 2b). A sensibility analysis of the quality of the fit (measure with $R^2$) between experimental points and the fitted polynomial surface was performed to choose the degree of the polynomial surface (supplementary material Figure S2): The polynomial surface chosen was of degree 5 in strain and of degree 1 in strain rate ($R^2 > 0.97$). To ensure that the surface crossed the 0 abscissa, the point of origin of each curve was weighted with the weighting of 10. Then, the stress vs. strain curves at four constant strain rates (50, 100, 150, and 200 s$^{-1}$) were extracted from the polynomial surface (Figure 2c).

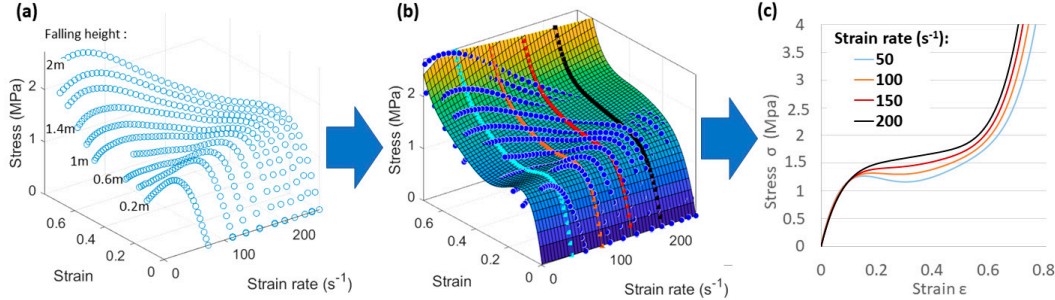

**Figure 2.** Method to obtain stress vs. strain curves at a constant strain rate: (**a**) Plotting the curves strain ($\varepsilon$), strain rate ($\tau$) and stress ($\sigma$) from different impact heights in a 3D space, (**b**) polynomial surface fitting, and (**c**) extraction of the four stress vs. strain curves.

The absorbed energy per unit volume and the efficiency diagrams of the VN foams were plotted to evaluate and compare their impact energy absorption. The absorbed energy per unit volume (W) corresponds to the area under stress vs. strain curve up to the point of interest (Equation (2)) [6]. The efficiency (Eff) is defined by Miltz and Gruenbaum [30] as the efficiency of energy absorption of a compressed foam compared with an ideal one that transmits the same but constant force when fully compressed. The efficiency of energy absorption is calculated as the ratio of absorbed energy up to the point of interest divided by the stress $\sigma_1$ at this point (Equation (3)) [5,31]. The energy per unit volume and the efficiency were plotted against the

stress, to identify the optimum usage of the foam in terms of energy absorbed and transmitted stress [5,30]. The stress, strain, and absorbed energy at maximum efficiency were calculated.

$$W = \int \sigma * d\varepsilon \tag{2}$$

$$Eff = \frac{\int_0^{\varepsilon 1} \sigma * d\varepsilon}{\sigma 1} \tag{3}$$

The stress vs. strain curves of VN foams displayed three regions: Linear elasticity, plateau, and densification. The stress and the strain at yield and at the onset of densification were identified from the experimental results, as well as the Young's modulus. The Young's modulus was measured as the slope of the curve at 0 strain. The yield stress was identified as the intersection of the tangents of the plateau stress region (from $\varepsilon = 0.1$ to $\varepsilon = 0.5$) and the elastic region [32]. The onset of densification was defined as the point of maximum efficiency, according to Li et al. [33].

## 3. Results

Engineering stress vs. strain curves are presented in Figures 3 and 4 for compression and combined compression and shear, respectively. The three foam materials displayed typical elastic, plateau, and densification phases in both types of loading. The surface fit accounted for over 90% of the total variation in the data for both impact compression and combined compression and shear. The experimental data, the coefficients of the fitted surfaces, and the surface plot for combined compression-shear loading are available in the supplementary material (Figure S3, Table S4, and Figure S5, respectively). The energy- absorption diagrams are presented in Figure 5 with efficiency vs. stress curves. Finally, Table 1 summarizes the key parameters of the stress vs. strain and energy-absorption curves for constant strain rates of 0.01 and 100 s$^{-1}$.

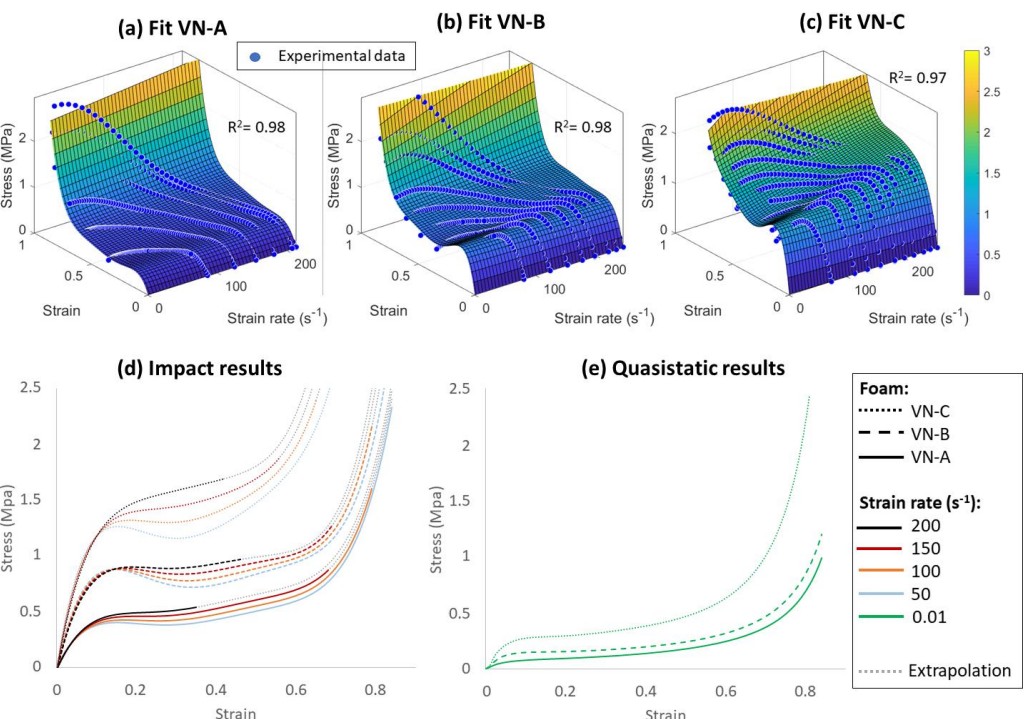

**Figure 3.** Strain rate dependent behavior of vinyl nitrile (V) foams in compression: (**a–c**) Surface plots and experimental data (markers) showing engineering stress vs. strain and strain rate at impact for (**a**) VN-A, (**b**)VN-B, and (**c**) VN-C; (**d**) fitted engineering stress vs. strain curves of the three VN foams at a strain rate of 50, 100, 150, and 200 s$^{-1}$; (**e**) engineering stress vs. strain curves of the three VN foams in quasi-static loading. The part of the curves that is extrapolated beyond the tested range is marked with gray dashed lines.

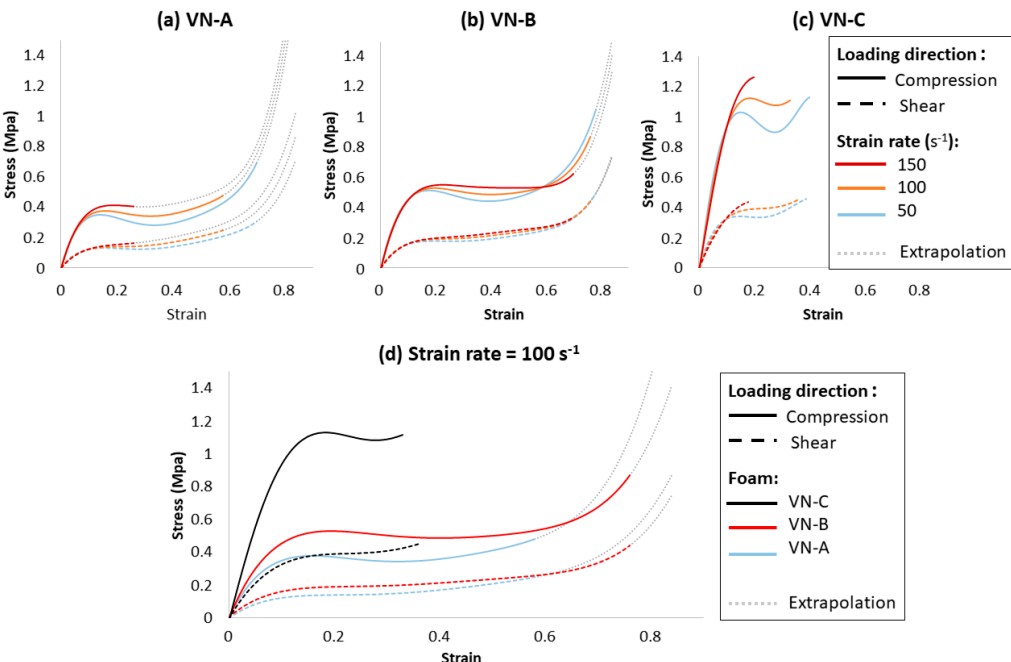

**Figure 4.** Strain rate dependent behavior of VN foams in combined compression and shear loading: (**a**–**c**) Fitted engineering stress vs. strain curves in compression and shear directions at a strain rate of 50, 100, and 150 s$^{-1}$ for (**a**) VN-A, (**b**) VN-B and (**c**) VN-C; (**d**) fitted engineering stress vs. strain curves in compression and shear directions for the three VN foams at 100 s$^{-1}$. The part of the curves that is extrapolated beyond the tested range is marked with gray dashed lines.

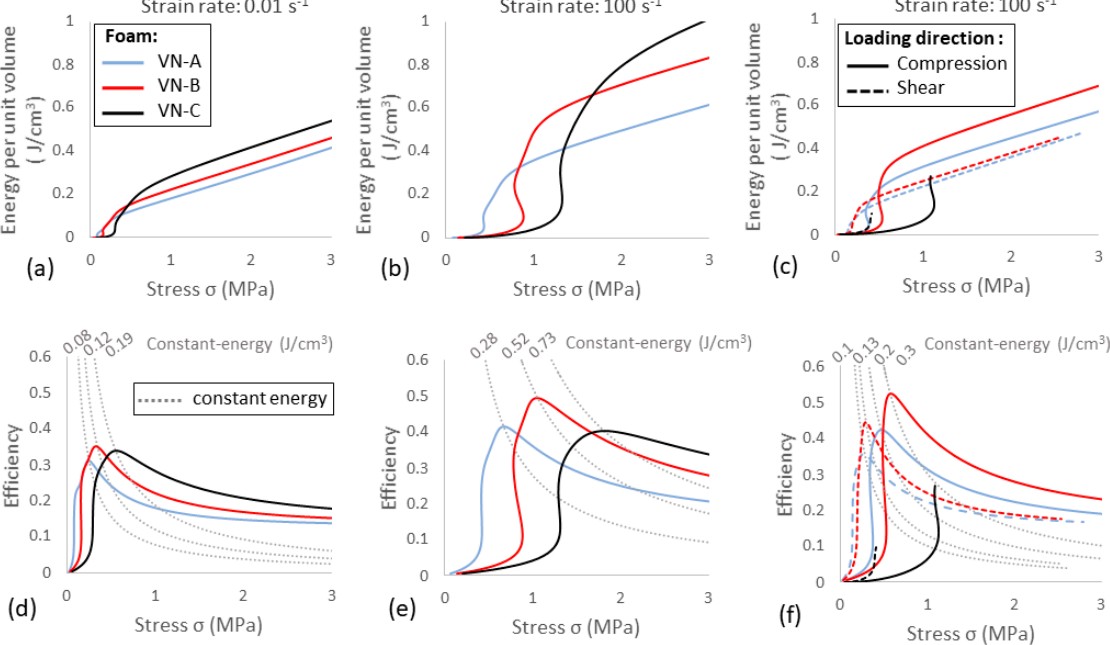

**Figure 5.** Energy-absorption diagrams of VN foams: (**a**–**c**) Energy per unit volume vs. engineering stress of the three VN foams, in (**a**) quasi-static compression (strain rate 0.01 s$^{-1}$) (**b**) impact compression (strain rate 100 s$^{-1}$) and (**c**) impact combined compression and shear loading (strain rate 100 s$^{-1}$); (**d**–**f**) efficiency vs. stress curves and constant-energy levels at maximum efficiency in (**d**) quasi-static compression (strain rate 0.01 s$^{-1}$), (**e**) impact compression (strain rate 100 s$^{-1}$) and (**f**) impact combined compression and shear loading (strain rate 100 s$^{-1}$).

**Table 1.** Parameters of the stress-stain curves of the three VN foams in quasi-static compression (strain rate 0.01 s⁻¹), impact compression, and impact combined compression and shear loading (strain rate 100 s⁻¹). Abbreviations: E* Young's modulus; $\varepsilon_{pl}$ and $\sigma_{pl}$ strain and stress at yield; $\varepsilon_{eff.}$ $\sigma_{eff.}$ And $W_{eff.}$ Strain, stress, and energy per unit volume at the point of maximum efficiency (at the onset of densification); $\sigma_s/\sigma_c$ ratio between the stress in shear and in compression at a given strain; $W_s/W_c$ ratio between the energy per unit volume in shear and in compression at a given strain.

| | E* (Mpa) | $\varepsilon_{pl}$ | $\sigma_{pl}$ (Mpa) | $\varepsilon_{eff.}$ | $\sigma_{eff.}$ (Mpa) | $W_{eff.}$ (J/cm³) | | | | |
|---|---|---|---|---|---|---|---|---|---|---|
| **Quasi-static Compression (0.01 s⁻¹)** | | | | | | | | | | |
| VN-A | 1.60 | 0.03 | 0.05 | 0.63 | 0.25 | 0.08 | | | | |
| VN-B | 2.89 | 0.04 | 0.12 | 0.62 | 0.33 | 0.12 | | | | |
| VN-C | 4.81 | 0.05 | 0.23 | 0.57 | 0.56 | 0.19 | | | | |
| **Impact Compression (100 s⁻¹)** | | | | | | | | | | |
| VN-A | 7.71 | 0.049 | 0.38 | 0.62 | 0.68 | 0.28 | | | | |
| VN-B | 16.30 | 0.052 | 0.85 | 0.64 | 1.05 | 0.52 | | | | |
| VN-C | 22.24 | 0.054 | 1.21 | 0.57 | 1.82 | 0.73 | | | | |
| **Impact Compression and Shear (100 s⁻¹)** | | | | | | | $\sigma_s/\sigma_c$ at $\varepsilon_{pl.}$ | $W_s/W_c$ at $\varepsilon_{pl.}$ | $\sigma_s/\sigma_c$ at eff. | $W_s/W_c$ at eff. |
| VN-A_comp. | 6.75 | 0.053 | 0.36 | 0.58 | 0.48 | 0.20 | 0.32 | 0.38 | 0.52 | 0.44 |
| VN-A_shear | 2.31 | 0.050 | 0.12 | 0.61 | 0.27 | 0.10 | | | | |
| VN-B_comp. | 8.02 | 0.066 | 0.53 | 0.64 | 0.58 | 0.30 | 0.31 | 0.37 | 0.49 | 0.41 |
| VN-B_shear | 2.83 | 0.058 | 0.16 | 0.65 | 0.29 | 0.13 | | | | |
| VN-C_comp. | 12.64 | 0.085 | 1.08 | NA | NA | NA | 0.32 | 0.36 | NA | NA |
| VN-C_shear | 5.61 | 0.062 | 0.35 | NA | NA | NA | | | | |

## 3.1. Effect of Density

Both Young's modulus and stress at the plateau increased with increasing foam density. The measured density was 97.5 kg/m³ for VN-A, 125 kg/m³ for VN-B, and 183 kg/m³ for VN-C. In quasi-static compression, the Young's modulus of VN-A samples was 2.9 and 4.9 times smaller than for VN-B and VN-C samples, respectively (Table 1). Similarly, the stress at the onset of densification of VN-A samples was 1.2 and 1.7 times smaller than for VN-B and VN-C samples, respectively. Consequently, the absorbed energy at maximum efficiency of VN-A samples was 1.5 and 2.5 times smaller than for VN-B and VN-C samples, respectively (Figure 5). The plateau length also varied between the three VN foams: The strain at the offset of densification was smaller for the VN-C samples (0.58) than for the VN-A (0.69) and VN-B (0.67) samples. Similar trends can be seen in impact compression and in combined compression-shear loading, in both the compression and shear directions (Figures 3 and 4). The maximum energy absorption efficiency was higher for VN-B samples (49%) than for VN-A (41%) and VN-C (40%) samples. Finally, the measured Poisson's ratio differed between the three foams, with values of 0.15 for the VN-A samples, 0.21 for the VN-B samples, and 0.18 for the VN-C samples.

## 3.2. Effect of the Strain Rate

Vinyl nitrile foams displayed strain rate dependent behavior. In compression, the stress at the onset of densification was about twice as high at 100 s⁻¹ than at 0.01 s⁻¹ for the VN-A and VN-B samples, and almost three times higher for the VN-C samples (Table 1). At maximum efficiency, the energy absorbed by the three VN foams was about four times higher at 100 s⁻¹ than at 0.1 s⁻¹, and the transmitted stress was about three times higher (Table 1 and Figure 5). Between 50 and 200 s⁻¹, the stress of the plateau also increased by about 1.4 times in the three VN foams. This increase in the stress of the plateau with strain rate was also seen in the combined loading, in both the compression and shear directions. Strain rate also affected the maximum energy absorption efficiency, with values of 35% in quasi-static and 51% at 100 s⁻¹ for the VN-B in compression.

### 3.3. Effect of the Loading Direction

In combined compression and shear (45° angle), the ratio between shear stress and compressive stress ($\sigma_s/\sigma_c$) was, respectively, equal to 0.32, 0.31 and 0.32 at yield ($\sigma_{pl.}$) for VN-A, VN-B, and VN-C at 100 s$^{-1}$ (Figure 4, Table 1). This ratio increased up to about 0.5 at maximum efficiency for both VN-A and VN-B. This information was not available for VN-C because the onset of densification was not reached at the maximum drop height. The compressive stress at yield, and the energy per unit volume at maximum efficiency were smaller in combined loading (0.43 MPa and 0.3 J/cm$^3$ for VN-B) than in pure compression (0.75 Mpa and 0.52 J/cm$^3$ for VN-B). The sum of compressive and shear energy at maximum efficiency was almost equal to that of compression for VN-A (0.3 and 0.28 J/cm$^3$), but was smaller for VN-B (0.43 and 0.52 J/cm$^3$).

## 4. Discussion

This is the first study to describe the strain rate dependent behavior of VN foam materials submitted to compression and combined compression and shear. To do so, a method to obtained stress vs. strain curves of foam materials at constant strain rates from a drop test was adapted [8]. This characterization includes stress vs. strain behavior, as well as energy efficiency, and can thus, be used to compare VN foams to other helmet liner materials and could help in the design of new protective devices.

### 4.1. Effect of the Strain Rate

Strain rate influenced the VN foams behavior and energy absorption capabilities: The absorbed energy was four times higher at 100 s$^{-1}$ than at 0.01 s$^{-1}$. This is not surprising as the strain rate dependency has been established for other polymer foams [5,8,22,34]. The difference between VN foams behavior at 100 s$^{-1}$ and at 0.01 s$^{-1}$ is probably due to a combination of the strain rate dependency of the base material [35], and both distinctive cell deformations [36], and different contributions to the gas pressurization in the closed cell between quasi-static and impact loading [24,32]. The first consequence of this result is that VN foams must be tested at strain rates relevant to the intended use. For instance, the transmitted force of VN-B at maximum efficiency was three times higher at 100 s$^{-1}$ than in quasi-static compression. Thus, candidate material selection should not be based solely on quasi-static material data. Secondly, the high strain rate dependency makes it difficult to precisely compare the results with foams from other studies. Indeed, high strain rate characterization of foams is often performed with drop tests where the impactor decelerates during the deformation of the sample [5,22,37,38]. For instance, in our experiments, with the same impact speed of 4 m.s$^{-1}$, the strain rate at 50% strain was 170 s$^{-1}$ for VN-A, 120 s$^{-1}$ for VN-B, and only 55 s$^{-1}$ for VN-C (Figure 3a–c). The strain rate variation also depends on the mass and the speed of the impactor and the dimensions of the foam samples. Thus, when comparing two stress vs. strain curves (or two energy diagrams) with the impact speed only, the authors compare stress vs. strain curves with different strain rates. In this article, the method of Krundeva et al. [8] was adapted. This method consisted of interpolating a stress vs. strain curve at constant rate between stress vs. strain curves from drop test results at several impact speed. However, instead of using linear interpolation between the experimental stress vs. strain curves, a polynomial surface fitting was used to obtain the equation of the fitted surface. The advantage of this polynomial fit is that no additional knowledge of the foam is necessary: It is a phenomenological model requiring only the displacement and the force recorded during the impact tests and accounting for over 90% of the total variation in the data. However, this method could also be implemented with other mathematical models specifically developed to represent foam behavior: Phenomenological models [39] or models accounting for foam density or foam structure [6,40]. In future works, similar methods could be used when characterizing and comparing strain rate dependent materials using drop tests.

### 4.2. Effect of Density

The stress vs. strain behavior of the three VN foam materials were similar in shape, and the Young's modulus and stress at the onset of densification increased with density in both compression and combined compression-shear loading even though the efficiency was not the same (Table 1). This is consistent with previous research reporting that the Young's modulus of various cellular foams can be represented by a quadratic function of the relative density [6]. Also, in combined loading, the ratio between shear and compressive stress ($\sigma_s/\sigma_c$) was similar between the three VN foams. Such consistent behaviors suggest that it may be possible to predict the behavior of other VN foams based on their density, as proposed for other polymeric foams [5]. Further work characterizing additional VN foam densities is needed to test that hypothesis.

### 4.3. Vinyl Nitrile Foam vs. Expanded Polypropylene Foam

The characteristics of the VN foams were compared to those of EPP foam, one of the most used and described foam for cushioning in helmet liners. In compression loading, the VN energy–absorption diagram and efficiency vs. stress curves at 100 s$^{-1}$ were compared to EPP results obtained by Avalle et al. [5] at 60 s$^{-1}$ initial strain rate. At equivalent density, EPP foam had a higher Young's modulus, higher plateau stress, and thus, absorbed more energy at maximum efficiency than the tested VN foam. For instance, at maximum efficiency, EPP106 ($\rho^* = 94.6$ kg/m$^3$, R = 0.10) absorbed more energy than VN-A with similar density ($\rho^* = 97.5$ kg/m$^3$, R = 0.09−0.11), but the same amount of energy as VN-B, which was denser ($\rho^* = 125$ kg/m$^3$, R = 0.09−0.11). However, VN-B had a higher efficiency than EPP106 (52% vs. 33%), and thus, transmitted less stress (1.05 vs. 1.57 MPa). Thus, to absorb the energy of 0.52 J/cm$^3$, VN would be a heavier, but more efficient choice than EPP. Similarly, the energy absorbed at maximum efficiency by VN-A ($\rho^* = 97.5$ kg/m$^3$, R = 0.09−0.11) was similar to that of EPP70 ($\rho^* = 60.5$ kg/m$^3$, R = 0.06) (0.28 vs. 0.34) with a better efficiency (41% vs. 36%). In combined compression and shear, the ratio between shear and compressive stress ($\sigma_s/\sigma_c$) in VN foams was compared to that in EPP foams measured by Mills et al. [20]. They impacted a 43 kg/m$^3$ (R = 0.5) EPP foam, at an initial rate of 120 s$^{-1}$ with an experimental setup similar to the one used here (45° impactor). Based on their stress vs. strain curves, the ratio between shear and compressive stress ($\sigma_s/\sigma_c$) in EPP was about 0.8 near maximum efficiency, which is larger than the ratio measured in VN-A and VN-B (0.52 and 0.49). Hence, for the same oblique impact, EPP transmitted proportionally less force in compression, but more in shear than VN. In this comparison, the VN foams tested were denser than the EPP foam, and it was assumed that, as in VN, the stress ratio ($\sigma_s/\sigma_c$) in EPP would not change substantially with density. The difference in the ratio ($\sigma_s/\sigma_c$) between EPP and VN might partly explain why Rousseau et al. [3] found that helmets with VN were better at reducing rotational accelerations than helmets with EPP, but worst at reducing linear acceleration. The differences observed between VN and EPP foam materials and the resulting differences in protection efficiency highlight the importance of testing foams in combined compression and shear, as combined loading is the most common loading for protective materials during an impact [9–12].

### 4.4. Use of the Results and Future Work

In this paper, the stress vs. strain behavior of VN foams is presented in the form of fitted polynomials. The coefficients of these polynomials are presented in the supplementary material Table S4 and can be used to predict the stress vs. strain curve of VN foams at any constant strain rate within the tested range. This can be useful to compare the behavior of VN foams with other foams at similar loading rates. These stress vs. strain curves could also be used to develop finite element models of VN foams. For instance, the stress vs. strain curves, as well as the Young's modulus, poison ratio, and density of VN foams can be used as inputs in tabulated strain rate dependent law for viscoelastic materials (i.e., law 38 in RADIOSS, Mat modified crushable foam in LS-Dyna) [41,42] which are widely used to model foams [8,43].

## 5. Conclusions

Three VN foams were characterized in compression and combined compression and shear at different strain rates. VN foams displayed strain rate dependent behavior, and the stress at plateau increased with the foam density. In compression, VN-A, the lighter foam (97.5 kg/m$^3$) absorbed 1.5–2.5 times less energy at maximum efficiency than VN-B (125 kg/m$^3$) and VN-C (183 kg/m$^3$), respectively. VN-B presented the highest maximal efficiency (49%) in both compression and combined loading. Compared to EPP foams, VN foams were denser for the same amount of energy absorbed, but presented higher maximal efficiency, reducing stress for the same amount of impact energy. Finally, when VN foam samples were submitted to combined loading, more than 65% of the stress was in the compression direction, which is more than in EPP, where the stress is more evenly distributed between compression and shear. These characteristics make VN foam an interesting candidate for energy absorption purposes, especially in helmets where increased shear force can induce increased rotational accelerations of the head. For this characterization, a methodology to obtained stress vs. strain curves at constant strain rates from a drop test was refined. This methodology enables a more consistent comparison between tested foam materials and could support finite element modeling in future works.

**Supplementary Materials:** The following are available online at http://www.mdpi.com/2076-3417/10/22/8286/s1, Figure S1: Example of full field strain measurement using digital image correlation, Figure S2: Sensibility analysis of the quality of the fit, Figure S3: Impact stress vs. strain data, Table S4: Coefficient of the equation of the fitted surface, Figure S5: Surface plot of the Strain rate dependent behavior of VN foams in combined compression and shear, Video S1: Quasistatic compression of VN foams.

**Author Contributions:** Conceptualization, N.B., Y.P. and E.W.; Funding acquisition, N.B., Y.P. and E.W.; Investigation, N.B., J.-M.D., O.L. and S.L.; Methodology, N.B., Y.P., J.-M.D. and E.W.; Supervision, Y.P., S.L. and E.W.; Writing—original draft, N.B.; Writing—review & editing, N.B., Y.P. and E.W. All authors have read and agreed to the published version of the manuscript.

**Funding:** This research was funded by the Natural Sciences and Engineering Research Concil of Canada and by the company CCM Hockey.

**Conflicts of Interest:** The study was partly funded by the company CCM Hockey which design, manufacture and sell helmets for hockey players, one of the authors was employed by CCM Hockey at the time of the study.

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
