# Peer review of "Strain Rate Dependent Behavior of Vinyl Nitrile Helmet Foam in Compression and Combined Compression and Shear"

_applsci, doi:10.3390/app10228286_

Round 1

Reviewer 1 Report

  1. This manuscript is a very good topic to describe the material of helmet and test them.
  2. The conclusion should be included the comparison of VN-A, VN-B and VN-C.
  3. How to produce the materials of VN-A, VN-B and VN-C!

Author Response

The authors would like to thank the reviewers for their remarks and suggestions, which enabled us to improve the paper significantly. All these comments were considered. Please find bellow the detailed responses to the reviewer’s comments.

Response to Reviewer 1 Comments

This manuscript is a very good topic to describe the material of helmet and test them.

  1. The conclusion should be included the comparison of VN-A, VN-B and VN-C.

Response 1: The key difference between the three foams were added in the conclusion using the following sentences (P.11, L.356) : “VN foams display a highly strain rate dependant behaviour and the stress at plateau increases with the foam density. In compression, VN-A, the lighter foam (97.5 kg/m3) absorbed 1.5 and 2.5 time less energy at maximum efficiency than VN-B (125 kg/m3) and VN-C (183 kg/m3) respectively. VN-B presented the highest maximal efficiency (49%) in both compression and combined loading.”

  1. How to produce the materials of VN-A, VN-B and VN-C!

Response 2: According to the manufacturer, the VN foam are made of Nitrile Butyl Rubber and Poly Vinyl Chloride (NBR-PVC) and were produced using a hot press foaming process. We were not able to obtain more details from the manufacturer. A sentence was added in the text to better describe the material production (P.2, L.76) : ‘The VN foams were made of Nitrile Butyl Rubber and Poly Vinyl Chloride (NBR-PVC) and were produced using a hot press foaming process. They were received in large foam sheets (500 x 500 x 20 mm3) and were cut into 60 x 60 x 20 mm3 rectangular samples.’

Reviewer 2 Report

The authors have conducted impact tests of the VN specimens, and deduced the stress-strain curves at various constant strain rates. Their method seems to be useful for estimating deformation behavior at high strain rates. However, there are several issues to be clarified before publication.

(P.2, L.74).
According to Ref. [6], deformation of foams is basically dominated by its relative density (rho*/rhos), which is a ratio of the density of foam to that of matrix solid. Then, please show relative density for the VN samples.

(P.3, L.96)
The authors have used adhesive tape for setting the foam. The reviewer is anxious about its effects on the stress-strain curves. How can you exclude the artificial effects?

(P.4, first paragraph)
The authors have combined several stress-strain curves for analysis. To do so, the time origin of the measurements (i.e. the load and position) should be precisely controlled to be the same. Please describe how these measurements are triggered to each other (e.g. time chart).

(P.4, L.166, Eq.(f))
The definition of W [J/m3] seems to be wrong. W should be given by the area below the stress-stain curves: the integration of sigma * d(epsilon).

(P.4, L.168, Eq.(g))
The definition of Eff seems to be wired, because this value has residual dimension. The divider in the left hand side seems to be sigma_1*epsion_1.

(P.5, L.179)
It has been established that the Young's modulus of various cellular foams is represented by a quadratic function of the relative density [6]. Do you still claim linear correlation (P.5, L.179)?

(Figure 5)
The authors have plotted the Efficiency against the stress. Why did you choose stress, not strain? The reviewer considers the strain dependence is simpler and easier for understanding, because the stress sometimes gives mutivalued functions.

(Table 1)
The line drawings in the inset seems to be too rough and inaccurate. (e.g. Young's modulus seems to be underestimated, the slope in the plateau region seems to be overestimated, and the onset of densification seems to be somewhat arbitrary.) Please fix the drawing, and describe in the text how to determine these lines excluding ambiguity and arbitrariness.

(P.9, L.285)
Discussion on difference between VN and EPP should be based on the relative density to exclude the effect of the matrix. Please revise this paragraph.

There are many typos especially in units. The manuscript should be more carefully checked before submission.
(P.2, L.74, 76) "3" for kg/m3 and mm3 should be superscript.
(Fig.2(c), L.222 and 223 in P.6, Table 1, Fig. B1) Mpa should be MPa.
(Fig. 5 (a2, b2 and c2)) j/cm3 should be J/cm3.
(Fig.2(c)) No unit for the strain rate.
(Appendix C) Show the units for the empirical equation.

Reviewer 3 Report

General comments
Well written manuscript. Need minor revisions.

Specific comments
Main drawback of this manuscript is lack of information regarding morphology of the foam: is it closed cell or open cell. This must be added since morphology of foam has profound influence on its mechanical properties.

In line #85 authors state that "Video footage was analyzed with the GOM Correlate (Gom, Braunschweig, Germany) Digital Image
Correlation software to obtain lateral and axial engineering strains.." Authors need to either give more details or cite appropriate references.

In line #142 authors state "Appendix 1" but the manuscript doesn't have Appendix 1.

Authors also compare performance of their VN foams with EPP foams (43kg/m3) which in opinion of this reviewer is inappropriate since the density of EPP is far lower than VN foams. Authors must justify this inappropriate comparison.

Author Response

The authors would like to thank the reviewers for their remarks and suggestions, which enabled us to improve the paper significantly. All these comments were considered. Please find bellow the detailed responses to the reviewer’s comments.

Response to Reviewer 3 Comments

General comments
Well written manuscript. Need minor revisions.

Specific comments

  1. Main drawback of this manuscript is lack of information regarding morphology of the foam: is it closed cell or open cell. This must be added since morphology of foam has profound influence on its mechanical properties.

Response 1: The foams were closed cell foams, this was added in the methods section as well as in the abstract and introduction. Also, we added the composition and general information about the foams. The section now reads (P.2, L.75): “Three vinyl nitrile (VN) closed cell foams of density 97.5 kg/m3, 125 kg/m3 and 183 kg/m3 were tested. These three foams are referred as VN-A, VN-B and VN-C, respectively. The VN foams were made of Nitrile Butyl Rubber and Poly Vinyl Chloride (NBR-PVC) and were produced using hot press foaming process.”

  1. In line #85 authors state that "Video footage was analyzed with the GOM Correlate (Gom, Braunschweig, Germany) Digital Image Correlation software to obtain lateral and axial engineering strains." Authors need to either give more details or cite appropriate references.

Response 2: The previous sentence was completed to better explain what we did and previous works using similar method were cited. Also, one example of the performed DIC analysis was added in the Appendix A. The section (P.2, L.89) now reads:

“The face of each sample with speckle pattern was filmed using a MotionBLITZ EoSens® mini camera (MIKROTRON, Unterschleissheim, Germany) (4 Hz) to allow for full field strain measurement using digital image correlation (DIC) as per previous work [26–28]. Video footage was analyzed with the GOM Correlate (Gom, Braunschweig, Germany) DIC software to obtain lateral and axial local strains (Appendix A).”

  1. In line #142 authors state "Appendix 1" but the manuscript doesn't have Appendix 1.

Response 3: Yes, sorry about that, this was a typo: we referred to Appendix B which can be found P.13.

  1. Authors also compare performance of their VN foams with EPP foams (43kg/m3) which in opinion of this reviewer is inappropriate since the density of EPP is far lower than VN foams. Authors must justify this inappropriate comparison.

Response 4: In this section, we initially compared our VN foams with EPP foams of similar density. The following sentence was added to makes it clearer (P.10, L.327): “At equivalent density, EPP foam presented a higher young modulus, a higher plateau stress and thus absorbed more energy at maximum efficiency than the tested foam. For instance, at maximum efficiency, EPP106 (ρ*= 94.6kg/m3) absorbed much more energy than VN-A with similar density (ρ*= 97.5 kg/m3) but the same amount of energy that VN-B, which was much heavier (ρ*= 125 kg/m3).”

However, regarding the combined compression and shear loading, only one experiment has been reported with EPP (Mills and Gilchrist 1999) and they only tested an EPP of 43kg/m3. We did not compare the absorbed energy nor the young modulus but only the ratio of the energy absorbed in compression and the energy absorbed in shear. The initial sentence of that comparison was modified to clarify and narrow the scope of the comparison (P.10, L337). It now reads : “In combined compression and shear, the ratio between shear and compressive stress (σs/σc) in VN foams was compared to that in EPP foams (43kg/m3) measured by Mills et al. [20].”

According to our experiments, in the VN foams, this ratio did not significantly vary with the density (always about 0.5). So we made the assumption that this ratio would not drastically change with density in other foams. That is why we compared this ratio with that of another foam even if the density of that foam was lower. A sentence was added to justify that comparison (P.10, L.344) : “In this comparison, the VN foams tested were denser than the EPP foam and it was assumed that, as in VN, the stress ratio (σs/σc) in EPP would not change significantly with density.”

Round 2

Reviewer 2 Report

The authors have appropriately revised the manuscript. The reviewer considers the revised version is worthy for publication. I would like to ask the authors to fix two typos in the revised part; "young modulus" should read as "Young's modulus" in L.305 and L.317 in P.10. 

Author Response

Response to Reviewer 2 Comments

1. The authors have appropriately revised the manuscript. The reviewer considers the revised version is worthy for publication. I would like to ask the authors to fix two typos in the revised part; "young modulus" should read as "Young's modulus" in L.305 and L.317 in P.10.

Response 1: Thank you for your review. The typo was fixed throughout the text.